

# Carbon and Nitrogen Dynamics in Subsoils After 20 years of Added Precipitation in a Mediterranean Grassland

Leila M. Wahab[1], Sora L. Kim[1], Asmeret A. Berhe[1]

[1]Life and Environmental Sciences, University of California - Merced, Merced, 95343, USA

*Correspondence to*: Leila M. Wahab (lwahab@ucmerced.edu)

**Abstract.** Precipitation is a major driver of ecosystem change and physiochemical characteristics of soil. Under different climate change scenarios, increased drought frequency and changing precipitation are predicted to impact Mediterranean ecosystems, including in Northern California. Studies based on two major climate models investigated the impact of increased precipitation in parts of California where the additional precipitation occurred in winter or spring months. It was

found that changing precipitation seasonality has significant impacts on plant community dynamics, microbial and fungal dynamics, and abiotic processes in soil. Subsoils are large carbon reservoirs. However, most studies investigating precipitation effects on soil organic matter (SOM) primarily focus on near-surface soils. Recent studies indicated different responses to environmental perturbation in surface (<30cm) versus deep soils (>30cm) due to important differences in physiochemical characteristics. Here, we present soil data at depth (~300cm) from a 20-year precipitation manipulation

experiment. We determined changes in total elemental concentration and stable isotope composition of soil C, N, $\delta^{13}C$, and $\delta^{15}N$ for ambient control vs. additional precipitation in the winter and spring months. The addition of winter precipitation resulted in the largest cumulative C stock (0-300cm), however there were no statistically significant changes in carbon stock throughout the depth profile. However, there was evidence for vertical translocation of carbon to deep soil layers, specifically of plant-derived carbon, with both winter and spring precipitation additions. The precipitation addition in winter

also resulted in the highest subsoil carbon stock compared to the control (ambient) and spring treatments. Overall, added winter precipitation led to the best conditions for carbon accumulation since the added precipitation coincides with lower temperatures and improved growing conditions at our field site. This study highlights the importance of timing of precipitation events, especially with regard to deep carbon stocks (>1m).

Key words: Precipitation Experiment, soil carbon, soil nitrogen, precipitation seasonality

# 1 Introduction

Deep soil ranging from 1-3m can account for 30 to 50% of the total soil profile carbon stock (Jobbágy and Jackson, 2000), but most published studies only sample to 50cm or shallower (Yost and Hartemink, 2020). As a result, considerable uncertainty is associated with estimations of deep global soil organic carbon (SOC) stocks. Current estimates of global SOC



stocks to 1m have converged to 1100-1500 Pg, but estimates to 3m depths are 2800 Pg (± 700 Pg) (Jackson et al., 2017).

There are many reasons for the lack of data in deep soils, including logistical and cost issues associated with sampling to such great depth. However, deep soils are an active carbon pool and not full accounted as a potential sink or source under current and future climate change. The limited number of deep soil studies show that subsoil carbon storage is affected by climate change, land use, and management change. For example, the introduction of switchgrass significantly increased subsoil carbon stocks (Slessarev et al., 2020) while intensive agricultural management of grasslands decreased carbon stocks

at 100cm depth in Great Britian (Ward et al., 2016). Land use change and its effects on subsoil carbon are also moderated by climate (Guo and Gifford, 2002). The transition from forest to pasture increased in soil carbon in ecosystems with a mean annual precipitation (MAP) between 2000-3000cm, but decreased soil carbon in ecosystems with less than 1000mm and greater than 3000mm MAP (Guo and Gifford, 2002). However, most deep SOC responses to global change can only be hypothesized due to the few manipulative experiments on deep soils (Hicks Pries et al., 2023). Given that subsoils can be

affected by climate and land use change, and act as a significant reservoir of stable carbon, more experimental studies are needed.

Deep soils have key physical, chemical, and biological features that make them significant reservoirs of carbon as well as a target for long term carbon sequestration. Physical characteristics of deep soils include a greater bulk density and finer texture that result in greater physical protection of carbon (Button et al., 2022; Plante et al., 2006) and lead to lesser

accessibility of soil carbon compounds for microbial communities at depth (Wilpiszeski et al., 2019) . In terms of biological properties, there is also lesser microbial biomass in subsoils compared to the surface, though there is evidence of microbial biomass that can be reactivated with the presence of deep roots and/or mechanical disruption. Chemical features of deep soils lead to greater protection of carbon, which includes greater surface area of mineral surfaces, such as Fe/Al oxyhydroxides that allow for protective and stabilizing associations for carbon compounds that would otherwise be quickly

decomposed (Kleber et al., 2005; Porras et al., 2017; Schmidt et al., 2011). When this deep carbon stabilization takes place, older radiocarbon ages of subsoil carbon is often observed which is interpreted as greater residence time (McFarlane et al., 2013; Rumpel, 2004; Rumpel & Kögel-Knabner, 2011; Sollins et al., 2009). This combination of physical, chemical, and biological features demonstrates the potential of deep soils to hold significant amounts of stabilized carbon.

Subsoils in grassland ecosystems are particularly important to consider due to their large global land area and their

ability to store large amounts of carbon belowground (Bai & Cotrufo, 2022; Berhe et al., 2012; Chou et al., 2008). This large source of belowground biomass is due to deeply rooting grasses, and it is estimated that sixty percent of grassland net primary productivity (NPP) is stored belowground and is more likely to be incorporated in soil organic matter (Jackson et al., 2017). This belowground NPP results in a large soil carbon stock, and a meta-analysis found that grasslands have approximately 43% of grassland carbon stock is stored from 1-3m (Jobbágy and Jackson, 2000). It has been estimated that

44% of the variability in SOC stock uncertainty is associated with spatial scale and soil profile depth, especially due to the lack of data at greater depths and this results in significant uncertainty when estimating grassland carbon stocks globally



(Maillard et al., 2017). Sampling deeper soils to understand stabilization mechanisms and destabilization processes of carbon at depths greater than 1m will be key for deep grassland soils to serve as a carbon source or sink under climate change conditions with changing moisture and temperature regimes.

We lack measurements of physiochemical characteristics not just deep soils, but also the effects of environmental perturbation on deep soils. However, several recent studies indicated different responses to environmental perturbation in surface (<30cm) versus deep soils (>30cm) (Berhe et al., 2008; Hicks Pries et al., 2017; Min et al., 2020, 2021). Key questions remain about whether deep soil carbon is vulnerable to climate change and dynamics of other key nutrients at depth, like nitrogen and phosphorus. These dynamics are important because decoupling of these nutrients from each other could impact and alter carbon cycling. For example, increasing aridity was found to decouple key nutrients like phosphorus

(Delgado-Baquerizo et al., 2013). Long term field experiments are needed to test hypotheses at the field scale about the impact of environmental perturbation on deep soils. There is still limited data on SOC concentrations with increased precipitation in grasslands (Bai and Cotrufo, 2022). A study that looked at 30 years of precipitation augmentation in a grassland ecosystem found minimal changes in bulk carbon or nitrogen, but did observe greater mineral associated organic matter (MAOM) in the top 30cm (Rocci et al., 2023). Overall, there are few long-term environmental manipulations and

even fewer have specifically examined the impacts of long-term manipulation on deep soils. This makes the Angelo manipulation experiment, which has been ongoing for 20 years, a particularly good site to ask questions regarding changes in soil biogeochemistry with decadal scale precipitation shifts. This site was had multiple studies occur at the 6 and 10 year mark of the experiment (Berhe et al., 2012; Cruz-Martínez et al., 2012; Hawkes et al., 2011; Suttle et al., 2007), but this

study represents one of the first long term follow ups on that experiment to great depth.

        Stable isotopes are a staple tool in the realm of soil science as ecological integrators, and can be a useful tool in understanding environmental perturbations in deep soils. C3 plants in particular have a well-documented physiological response to increasing aridity that leads to high $\delta^{13}C$ values (Farquhar et al., 1989; Kohn, 2010). $\delta^{13}C$ has been shown to vary with mean annual precipitation (MAP) due to discrimination against $^{13}C$ in drier areas because stomata have to remain

close more to minimize water loss (Krüger et al., 2023). This interaction between precipitation and the stable isotope values of plant matter means that the inputs for formed carbon will be affected by climate. These altered stable isotope values could act as a potential tracer for plant inputs into soil. There is a well-documented pattern of increasing $\delta^{13}C$ values with depth; a disruption of this pattern by $^{13}C$ depleted plant matter inputs could be a good indicator of formed carbon being distributed throughout the profile. Furthermore, Diffuse Reflectance Mid-Infrared Fourier Transform spectroscopy (DRIFTS) is a

complementary analysis that can characterize the chemical composition of soil carbon. DRIFTS can measure the vibrational frequency of ecological relevant functional groups, like aliphatic, aromatic, and amide functional groups (Mainka et al., 2021; Margenot et al., 2015; Parikh et al., 2014). Together, stable isotopes and DRIFTS can provide important information about incoming plant matter as well as microbial activity in soils.



We wanted to examine the intersection of a long-term precipitation manipulation experiment and its potential
impacts on deep soils in a northern California grassland to understand whether subsoil carbon stocks might be affected by
climate change. This experiment has been ongoing for 20 years, and is testing the impacts of increased precipitation
combined with changing seasonality. More specifically, it is testing the impact of shifting seasonality of precipitation to the
spring months (Mar-June) in a Mediterranean climate where most of the precipitation for the water year typically takes place
in the winter months (Nov-Feb). The objectives of this study were to determine how changes in the amount and timing of
rainfall in a California grassland ecosystem affect: (a) the distribution of carbon stocks to 3m, (b) the chemical composition
of organic matter entering soil and its distribution throughout the soil profile, and (c) the associations between inputs and
carbon stocks.

**2.1 Site Description**

The Angelo Precipitation Experiment was established in a meadow at Angelo Coast Range Reserve in Mendocino
Country, California (39° 44' 21.9762" N, -123° 37' 50.8722" W). The dominant vegetation at Angelo is a mix of *Aira* spp.,
*Bromus* spp, and *Briza* spp. (Foley et al., 2023). The site is at an elevation of 1,350 m.a.s.l and experiences a Mediterranean
climate with wet, cool winters and warm, dry summers. At Angelo, soils are part of the Holohan-Hollowtree-Casabonne
Complex, are classified as Ultic Haploxeralfs. The parent material is largely graywacke and mudstone, and derived from
Cretaceous marine grey-wacke sandstones and mudstones of the Franciscan complex (Berhe et al., 2012).

**2.2 The Rainfall Addition**

The Angelo Reserve rainfall manipulation experiment was established in 2000 and set up to reflect changes in
rainfall patterns predicted for Northern California over the next 50-100 years by the Hadley Centre for Climate Prediction
and Research (HadCM2) and the Canadian Centre for Climate Modeling and Analysis (CCM1).  For this experiment, thirty-
six large circular plots (70-m$^2$) were regularly spaced across 2.7 ha meadows. Plots were set up in random block design for
three separate treatments: (1) an ambient rainfall control; (2) a winter-addition of precipitation; (3) a spring addition of
precipitation. Water addition treatments were administered by adding 14-16 mm of water every third day over three months
(Fig.1). For the winter treatment, this water addition was administered from January to March, and for the spring treatment,
it was administered from April to June (Fig.1). This water addition results in a 20% increase over the mean annual
precipitation (Suttle et al., 2007). The supplemental water for the water addition experiment was collected from a spring
above the meadow and distributed evenly over the surface of each plot through a sprinkler system (Rainbird ©
Raincurtain$^{TM}$).

Plots were assigned treatments in a randomized complete block design to take spatial biases into account (Suttle et
al., 2007). Within each experimental block, treatment assignment is randomized among the plots, and then re-randomized for
the next block and so on. This results in each block containing a single replicate of each experimental treatment. This design



maximized the likelihood of any pre-existing differences that might exist in terms of physical or biological conditions across the grassland.

### 2.3 Soil Sampling

In October 2020, samples were collected by Geoprobe to depth of resistance (approximately 3m) with 4 replicates
per treatment (Ambient, Winter, and Spring). For all cores, samples were collected at consistent 10cm intervals (0-10, 10-20, and so on).

After samples were collected, they were transported in coolers with ice packs and stored in a 4℃ cold room for approximately 4 months until they could be subsampled and analyzed. Long storage times occurred due to a lack of access to laboratory facilities due to the COVID-19 pandemic and subsequent shutdown procedures. When samples could be
processed, a subsample was removed from each sample, and air dried for 7 days at room temperature. Soil samples were tested for carbonates by observing the presence and degree of effervescence with a few drops of 1 M Hydrochloric acid. Following air drying, the sample was then sieved to 2mm. A further subsample from the processed air-dried sample was taken for ball milling (using a Sample Prep 8000M Ball Mill) to a homogenous particle size.

We measured a suite of physical and chemical properties of these samples, specifically bulk density, soil pH, and
gravimetric water content. We collected bulk density at Angelo through Geoprobe cores and calculated carbon stocks with these bulk density estimates. We subsampled each depth increment to estimate water content, and then calculated the dry mass of soil in a 10cm increment. Bulk density was calculated as the mass of the dry >2mm fraction to correct for the impact of rock and root volume on soil carbon and nitrogen stocks (Throop et al., 2012). Soil pH was measured in a 1:1 soil:water and soil: $CaCl_2$ slurries.

### 2.4 Elemental and Isotopic Analyses

For elemental and isotopic analysis of C and N, soil samples were air dried, sieve to 2mm, and ground (using both a mortar and pestle and Sample Prep 8000M Ball Mill). The $\delta^{13}C$ and $\delta^{15}N$ values and elemental carbon and nitrogen contents of all samples were measured in the Stable Isotope Ecosystem Laboratory of (SIELO) the University of California, Merced. Briefly, samples were weighed into tin capsules and combusted in a Costech 4010 Elemental Analyzer coupled with a Delta
V Plus Continuous Flow Isotope Ratio Mass Spectrometer. Carbon and nitrogen isotope compositions were corrected for instrumental drift, mass linearity, and standardized to the international VPDB ($\delta^{13}C$) and AIR ($\delta^{15}N$) scales using the USGS 41A and USGS 40 standard reference materials. Mean $\delta^{13}C$ values for USGS 40 and 41a were (mean ± standard deviation with n indicated) -26.4 ± 0.1‰ (n = 118) and 36.5 ±0.2‰ (n = 59), respectively, and mean $\delta^{15}N$ values were -4.5 ± 0.1‰ (n = 118) and 47.5 ± 0.1‰ (n = 59), respectively. Elemental carbon and nitrogen content were determined via linear regression



of $CO_2$ and $N_2$ sample gas peak areas against the known carbon and nitrogen contents of USGS 40, USGS 41a, and Costech acetanilide. All isotope compositions are expressed in standard delta notations.

## 2.5 Diffuse Reflectance Infrared Fourier Transform Spectroscopy (DRIFTS)

To measure the presence of functional groups that are important for organic matter and mineral surfaces related to soil carbon across our study systems, we used diffuse reflectance mid-infrared Fourier Transform spectroscopy (DRIFTS).
DRIFTS measures the vibrational frequencies of functional groups in a sample, and is well suited for analyzing soils due to the minimal sample preparation needed for this technique. We performed analyses on bulk soil samples that were ball milled to a homogenous consistency to avoid interferences that could affect baselines or peak widths. We used a Bruker IFS 66v/S Spectrophotometer (Ettlingen, Germany) with a praying Mantis apparatus (Harrick Scientific, Ossining, NY) at the Nuclear Magnetic Resonance (NMR) lab at UC Merced. It is important to note that potassium bromide (KBr) was used as a
background reference, but samples were not diluted with KBr. Samples were initially dried in a desiccator following homogenization to remove interference from water. Absorption was measured between 4000 and 400 $cm^{-1}$ averaged over 300 scans with an aperture of 4mm. Functional groups for simple plant carbon (aliphatic C-H; λ: 2976-2898 $cm^{-}$), complex plant carbon (aromatic C=C; λ: 1550-1500 $cm^{-1}$), microbially derived carbon (amide/quinone/ketone, CO; aromatic, CC, carboxylate COO; λ: 1660-1580 $cm^{-1}$) were assigned following Mainka et al. (2022), also shown in Table 1 (Mainka et al.,
2021; Parikh et al., 2014; Vranova et al., 2013). We excluded wavenumbers that overlap with signal from mineral compounds, specifically from 1400-400 $cm^{-1}$, from the analysis (Margenot et al., 2015; Parikh et al., 2014). We also calculated ratios of simple plant carbon to microbial carbon, as well as complex plant carbon to microbial carbon by integrating the area under the curve. A low ratio of simple plant carbon to microbial carbon indicates microbial oxidation of plant derived carbon, and a high ratio of simple plant carbon to microbial carbon indicates a high supply of aliphatic plant
carbon to soil. Additionally, a low ratio of complex plant carbon to microbial carbon indicated more microbial oxidation of plant carbon, while a high ratio of complex plant carbon to microbial carbon indicates a high supply of aromatic plant compounds to soil.

## 2.6 Statistical Methods and Model Fitting

All statistical analyses were performed in R. Differences between treatments were evaluated through Kruskal-
Wallis test within each 10cm or 50cm depth interval depending on the analysis. We used a Kruskal-Wallis test combined with a Pairwise Wilcox due to it being a non-parametric statistical test. We determined that our data was non-parametric through a Shapiro Wilk test of normality. Statistical significance was evaluated using $\alpha = 0.05$, and all analyses were performed in R.

To test the relative importance of biotic and abiotic factors on $\delta^{15}N$ and carbon within treatments and to account for
possible nonlinear relationships, we used a hierarchical generalized additive mixed model (GAMM) (Pedersen et al., 2019).





To fit the GAMM we used the "mgcv" package (Wood, 2017). GAMMs are a type of generalized linear model where the predictor is defined by a number of smooth functions of covariates. Models avoid overfitting by penalizing each smooth function, or in other words, penalizing the "wiggliness" of the fit. We fit with smooth functions based on thin plate regression splines (which are the default) and residuals approximated a "scat" or scaled t distribution, which was assessed

from residuals using the package DHARMa (Hartig, 2022).  In order to compare the relative importance of abiotic (depth) and biotic (DRIFTS ratios) variables, we constructed different models: one model had all terms included, one only included depth, and one only included our DRIFTS ratios of interest that indicate microbial oxidation and plant inputs. All models were compared through the Akaike Information Criterion (AIC), and the full model with all predictors and our depth model had the lowest AIC, so not all of the models run were included for parsimony.

**3 Results**

**3.1 Variation in physical, chemical, and isotopic parameters across treatments and depths**

Physical parameters such as bulk density was relatively similar across treatments (Fig. 2a-b) though there were important differences in the surface. The control treatment had the highest bulk density at 10cm while the spring and winter treatments had lower bulk density (Fig. 2a). Bulk density increased across the depth profile, and eventually converged to similar values across all treatments (Fig. 2a). C (%) was greatest at the surface (0-30cm) in the control and spring treatments, but quickly

dropped off to similar values at approximately 50cm (Fig. 2b). N (%) was present in low concentrations across the entire profile and across treatments (Fig. 2c). However, C:N varied in key ways throughout the depth profile and between treatments (Fig. 2d). While C:N was the greatest at 10cm compared to the winter and spring treatments, the winter and spring treatments had elevated C:N from 30-100cm (Fig. 2d). Changes in physical and elemental parameters were largely

limited to the surface, and converged past 1m in most cases.

Stable isotope values to 3m were highly variable across the depth profile and across treatments, especially around the 100 cm depth. We did not observe treatment effects from added precipitation on $\delta^{13}C$ and $\delta^{15}N$ values (Fig. 2e-f). We expected the well-documented pattern of increasing $\delta^{13}C$ values with depth in soils (Natelhoffer and Fry, 1988) to be moderated by decreased $\delta^{13}C$ of formed C due to the added precipitation from the manipulation experiment in both the

winter and spring treatments. We observed slight differences in the overall distribution of $\delta^{13}C$ values in the winter and spring treatment throughout the depth profile.  However, except for the low $\delta^{13}C$ values we recorded around 300 cm depth in the spring treatment plots, we did not observe any statistically significant differences in $\delta^{13}C$ values throughout the profile.

In terms of chemical parameters, differences in pH between treatments were also limited to the top 1m and converged to similar values from 1-3m (Fig. 2g-2h). However, it is important to note that the spring and winter treatments

had higher pH ($H_2O$) compared to the control treatment to 1m. Only the spring treatment had elevated pH ($CaCl_2$), while the





control and winter treatments were similar throughout the entire depth profile. The pH values in water and $CaCl_2$ indicate slightly acidic to neutral soil pH across the soil profile for all treatments.

**3.2 Carbon stocks and elemental relationships across depth increments**

We found no statistically significant differences in carbon stock across the 3m depth profile or at the surface (top 50cm) from our Kruskal-Wallis test. All discussion of these results in this paragraph refer to gross changes (cumulative carbon stocks) or trends across the depth profile. We determined changes in the overall carbon stock across the entire depth profile of the treatment plots and observed important changes in seasonality. We found that the Winter treatment had the greatest carbon stock (200.5± 34.5 $g/cm^2$, table 2), followed by the control treatment (191.2±36.7 $g/cm^2$), while the spring had the smallest cumulative carbon stocks (171.4±13.7 $g/cm^2$) over the entire profile (0-300cm) (table 1). The soil carbon

stocks sharply dropped below 50cm in all the treatment plots. Proportionally, soils from 1-3m held 35% of the carbon stock in the control, 33% in the winter treatment, and 37% in the spring treatment (Fig. 3). In addition, a more detailed investigation of the surface revealed greater carbon stocks from 0-50cm in both the Winter and control treatments (Fig. 3). There was also relatively greater carbon in the winter and spring treatments at 150cm. Interestingly, the winter carbon stock at 300cm was decreased compared to both the control and spring treatments. Overall, however, the winter treatment had the

greatest gross carbon stock while the spring treatment had the least.

To further interrogate relationships between soil carbon processes and inputs at the precipitation manipulation experiment, we looked at relationships between carbon stock and C:N across depth increments (Fig. 4a-b). We expected to see significant positive relationships, indicating a tight relationship between carbon stock and plant inputs. We did indeed see positive and significant relationships across all treatments. We found a slightly more positive relationship in the winter

treatment than the control. In the Spring treatment, we saw a much narrower range in C:N values compared to the winter and control treatments. This analysis highlighted the unique distribution of both C:N and carbon stock in the spring treatment.

**3.3 Variation in functional group chemistry across treatments**

The DRIFTS spectra show differences across treatments and depths, especially areas of interest for biological OM inputs (Fig. 5). integrated for our analysis.  We observed shifts in functional group chemistry of SOM across treatments, both

proportionally (fig. 6) and in relationships to elemental and isotopic data (Fig 7). An important observation from the DRIFTS spectra is the proportional contribution of microbially associated functional groups across all three treatments (>50%; Fig. 6a-c). Simple plant-derived organic matter (aliphatics) was approximately 30% of the total, with some slight variations while complex plant matter (aromatic) functional groups were the smallest fraction, representing about 20% of the total (Fig. 6). Furthermore, differences in simple and plant-derived OM diverged between treatments from 200-250cm, where the control

treatment seemed to have the greatest aromatic or complex plant-derived OM proportionally (fig. 6a) and also based on the





complex: microbial ratio (fig. 6e). Our DRIFTS data overall suggested that the dominant functional groups were similar across depth and treatment, and that it was largely microbially associated organic matter.

We further analyzed the relationship of the shifts in SOM functional group chemistry to elemental and isotopic data to better understand abiotic versus biotic controls on SOM storage and processing across treatments. We chose to more
closely examine $\delta^{15}N$ because of it being coupled to biotic processes (Dijkstra et al., 2008; Hobbie and Ouimette, 2009), and C (%) to potentially understand relationships of biotic and abiotic factors with carbon storage. While a simple linear model did reveal some differences between treatments, the relatively poor fit of these models indicated possible nonlinear relationships between our predictors and ratios of interest (Figure S1).  Looking at the ratio of simple plant: microbial to $\delta^{15}N$ across treatments and depths, we observed an overall negative trend in the winter treatment, and a slightly positive and
significant ($p< 0.01$) trend for the spring treatment (Fig. S1a). In contrast, the ratio of complex plant: microbial to $\delta^{15}N$, there was a slightly negative and significant ($p< 0.002$) trend for the spring treatment only (Fig. S1b). To better represent these, we also fit hierarchical GAMMs to better predict both $\delta^{15}N$ (table 3a) and carbon concentrations (table 3b). GAMMS were fit to try and predict $\delta^{15}N$ and C(%) to better understand if  $\delta^{15}N$ is actually related to our DRIFTS ratios of interest (which we interpreted as biotic factors) and carbon storage. Through visualizing our GAMM, we saw unique relationships in the winter
treatment for the relationship between the simple plant: microbial and $\delta^{15}N$ values (Fig. S2a). Overall, we were able to account for greater variation for both $\delta^{15}N$ and C(%) using GAMMs. The model including all factors for C(%) performed better ($R^2_{adj} = 0.722$) than the model for $\delta^{15}N$ values ($R^2_{adj} = 0.133$). For both carbon concentration and $\delta^{15}N$, depth was a key factor across all treatments. Importantly, however, the simple plant: microbial DRIFTS ratio was a significant predictor of $\delta^{15}N$ for the winter treatment, and the complex plant:microbial DRIFTS ratio was a significant predictor for both the winter
and spring (Table 3a).

## 4 Discussion

Carbon translocation was a key mechanism in this experiment, and there was a clear impact on the distribution of carbon stocks and functional groups chemistry between topsoil and subsoil from adding more water to the soil profile. We saw the greatest cumulative carbon stocks in the winter treatment, especially compared to the spring addition. Furthermore,
plant phenology in an annual grassland is varies greatly from the winter months to the spring. Spring temperatures are increasing, evaporative stress is greater, and annual plants are usually reaching the end of their life span in this season. We think that greater carbon is accumulating in the winter treatment due to carbon addition coinciding with lower temperatures and lesser microbial activity. In sum, what we are likely seeing is increased plant inputs in the winter treatment and greater transport to deeper layers, whereas in the spring, due to higher temperatures and greater evaporative stress, there is more
gaseous loss of carbon, less water, and less carbon moved to deeper soil layers.





## 4.1 Carbon translocation and accrual as a result of added precipitation

We observed clear signals of greater carbon translocation throughout the soil profile as a result of added precipitation, but where this carbon accumulated was based on seasonality. Added precipitation seemed to lead to greater transport of carbon throughout the profile in the winter treatment especially. However, we did not see any statistically significant differences in
carbon stocks across both the subsoil and surface carbon stock measurements (Figure 3a-3b). This could be due to significant variability introduced from fixed depth bulk density measurements (von Haden et al., 2020). Previous work suggests that the main sources of organic matter into subsoils are plant derived compounds (roots and root exudates), dissolved organic matter (DOM) and bioturbation (Rumpel and Kögel-Knabner, 2011), and DOM and plant derived compounds are likely the dominant inputs to subsoils at Angelo. Work at the Angelo experiment has shown evidence of changing rooting patterns and
greater overall biomass with increased precipitation at this site (Suttle et al., 2007). A recent meta-analysis noted that decreased precipitation slows the belowground carbon cycle, while increases promote nearly every aspect, such as carbon stock, substrate supply, microbial activity, and respiration (Abbasi et al., 2020). This is due to interactions between precipitation and biological entities, namely plants and microbes. Increased precipitation root respiration and belowground NPP are positively correlated with soil water availability, and enhance plant growth and photosynthetic rates (Heisler-White
et al., 2008; Maire et al., 2015). Wetting of dry soil also has a dramatic impact on soil microbes due to increased substrate availability and reactivation of dormant microbes, yielding respiration pulses known as the Birch effect (Salazar et al., 2018; Schimel et al., 2007; Skopp et al., 1990). Overall, greater precipitation in the spring contributes to greater root exudation in surface soils that then gets quickly fixed by soil microbes. Our results show that in the winter treatment plots, the additional precipitation in the already wet winter season likely increases root exudation, where this increased carbon input coincides
with lower temperatures and lesser biological activity in soil. Thus, there is greater movement of this plant derived inputs moving down the profile, likely as DOM. Recent plot-scale studies have proposed that OM formation in subsoils is linked to a complex cascade model, in which OM is sorbed, microbially processed, and remobilized in cycles as it migrates down the profile (Liebmann et al., 2020). We observed significant evidence for this added carbon being plant derived based on both C:N and DRIFTS data. We also saw evidence of greater biotic processing in the Spring treatment when comparing $\delta^{15}N$ to
the simple plant matter: microbially associated DRIFTS ratio (Table 3). We also saw positive and significant trends when we related carbon concentrations to the ratio of complex plant matter to microbial associated OM (Fig. S2b), highlighting a unique relationship between complex plant inputs and carbon especially in the winter and spring that may be related to plant inputs. It is important to note that evidence of subsoil carbon accumulation in the winter treatment was only made possible by incorporating measurements from deeper than 1m, and that depth was a significant predictor in our GAMM models
across all treatments (Table 3).



## 4.2 Biotic shifts as a result of changing precipitation amount and seasonality

Changing plant phenology throughout the growing season and changes in plant community composition could be contributing to the differences we see in carbon stock accrual in our treatment plots. Increased precipitation is shown to increase net primary productivity in grasslands, but alter plant community composition and reduce diversity (Song et al.,
2019; Suttle et al., 2007). Furthermore, higher plant species richness is associated with increased soil organic carbon (Prommer et al., 2020). At the Angelo precipitation experiment, it was shown that plant community composition responses to changing precipitation were based on seasonality, with spring addition resulting in reduced plant diversity while the winter treatment maintained diversity close to the control (ambient) plots (Suttle et al., 2007). However, the spring treatment in our experiment still accumulated carbon in surface soils. This could be because invasive annual plants, such as cheatgrass, have
been shown to accumulate both carbon and nitrogen due to higher rates of root exudation according to mesocosm experiments (Morris et al., 2016). A recent meta-analysis further examined the feedback between annuals and litter and rhizosphere inputs, and found that invasive plants may support more decomposers that stimulate more nutrient release from litter (Zhang et al., 2019). The simplification of the plant community in the spring treatment plots as well as potential for greater root exudates and greater stimulation of decomposition by annual grasses at the surface could be leading to the
surface carbon stock accumulation, we observed from 0-50cm in the winter and spring plots. Other studies have suggested that a longer and later wet season would result in significant losses of carbon due to increased soil respiration (Chou et al., 2008). Increases in NPP accompanied with increased gaseous carbon flux is consistent with what is found in larger meta-analyses, where increased precipitation stimulates plant growth and ecosystem carbon fluxes (Wu et al., 2011). There is still a question at the Angelo precipitation experiment of how these changes in plant community will interact with potential
stimulation of gaseous carbon flux.

Isotope values can also be affected by interactions between plants and climatic conditions, specifically due to plant physiology responses to soil water conditions. The $\delta^{13}C$ value of C3 plants is tied to climatic regime (Krüger et al., 2023), furthermore the plants in this ecosystem are largely C3, meaning that differences in plant isotope values and formed carbon are likely driven by physiological responses to changing climatic regime and soil water.   More specifically, there is less
discrimination against the heavy isotope when plant stomata have to close more often, which is the case in arid environments (Casson and Gray, 2008; Driesen et al., 2020; Farquhar et al., 1989; Kohn, 2010; Krüger et al., 2023; Madhavan et al., 1991). Overall, this means that greater precipitation would drive $\delta^{13}C$ values down (more negative). There is also evidence that plant stomata opening is driven by soil water potential (Carminati and Javaux, 2020). Specifically, we think that formed carbon in plant inputs is lowered in $\delta^{13}C$ value, and is could be driving down $\delta^{13}C$ values in the soil profile. However, we
lack these measurements and think it would be a valuable avenue for future work.

Although plant communities are sensitive to changing precipitation regimes, microbes are resilient to precipitation shifts. Specifically, there is evidence that microbial community structure are resilient to long term shifts in precipitation seasonality, even if there are shifts in plant community structure (Cruz-Martínez et al., 2009). This capacity is likely because



the climatic history of mediterranean ecosystems would select for microbial populations resilient to soil moisture
fluctuations (Cruz-Martínez et al., 2009). While microbial communities are resilient to long term changes in moisture regime, there is evidence that they can respond rapidly to immediate changes in environmental conditions, which may be missed in long term studies (Cruz-Martínez et al., 2012). For example, subsoil microbial communities quickly respond to added carbon (Min et al., 2021) and old carbon can be quickly mineralized (Fontaine et al., 2007). These studies suggest that increases in carbon translocation to subsoils could stimulate the loss of ancient buried carbon through a potential priming
effect, where additions of fresh carbon can enhance the decomposition of harder to decompose or mineral associating carbon (Keiluweit et al., 2015; Kuzyakov et al., 2000).

**4.3 Implications for carbon sequestration potential of deep soils in grasslands**

While our results suggest possible carbon accrual in subsoils with winter addition of precipitation, it is important to consider mechanisms for destabilization of subsoil carbon. Addition of fresh carbon to subsoils is identified as a potential
destabilization mechanism due to priming effects (Rumpel and Kögel-Knabner, 2011). The addition of greater plant derived inputs in subsoils in the winter treatment could fundamentally alter carbon cycling in subsoils. Greater work is still needed on what proportion of added carbon can become mineral associated in subsoils or is quickly mineralized by soil microbes. Current evidence suggests that fresh carbon is quickly mineralized at depth (Fontaine et al., 2007), but few studies have looked at fresh carbon partitioning to the mineral associated fraction in subsoils. This added carbon could also be affecting
microbial community structure as well as increase the formation of necromass at depth. There is also evidence for changing porosity and soil structure to impact the structure of microbial communities in soils (Wilpiszeski et al., 2019). Overall, interactions between changing soil water conditions and carbon addition to subsoils is dependent on the seasonality of this added precipitation in a Mediterranean grassland, and could affect the sequestration potential of subsoils in grasslands under climate change.

**5 Conclusion**

This study leveraged a long term (20 years) precipitation manipulation experiment to investigate how changing precipitation amount and seasonality would affect soil carbon, nitrogen, and functional group chemistry in deep soils of a California grassland. We measured a suite of soil chemical characteristics, stable isotopes, carbon stocks, and performed Diffuse Reflectance Infrared Fourier transform Spectroscopy (DRIFTS) on all samples at 10cm increments for 0 to 3m and
found greater cumulative carbon stocks in the winter treatment. Across all treatments, we found that soils from 1-3 m held nearly a third of the overall carbon stock. These results suggest that added precipitation over the winter in Mediterranean grasslands can alter plant inputs and enhance carbon stocks in deep soils. Overall, this study highlights the importance of measuring soil carbon and functional group chemistry to greater depths.




**Author Contributions**

Conceptualization of the project led by AAB. Field and lab work was carried out by LMW. Formal analysis and investigation done by LMW with supervision and contributions from all authors. LMW prepared the manuscript with contributions from all co-authors.

**Acknowledgements**

We thank Debbo Elias, Charlotte Calvario, Todd Longbottom, Ronnie Hall, Xin Gao, and Jing Yan for assistance in field sampling and lab analyses. In addition, this study would not be possible without the field support from John Bailey at Hopland Research and Extension Center, Peter Steel from Angelo Coast Range Reserve, and Kate McCurdy at Sedgwick reserve, especially in August 2020 during the COVID-19 pandemic. This research was supported by the Department of
Energy, Sigma Xi Grants in Aid of Research, the UC Natural Reserve System Mathias Grant, and a UC Merced Committee on Research Senate Award.

**Funding**

This research was supported by the Department of Energy, Sigma Xi Grants in Aid of Research, the UC Natural Reserve
System Mathias Grant, and a UC Merced Committee on Research Senate Award. This research was also supported by the U.S. Department of Energy, Office of Biological and Environmental Research, Genomic Science Program 'Microbes Persist' Scientific Focus Area (#SCW1632) at Lawrence Livermore National Laboratory (LLNL).

**Competing Interests**

The authors declare that they have no conflict of interest



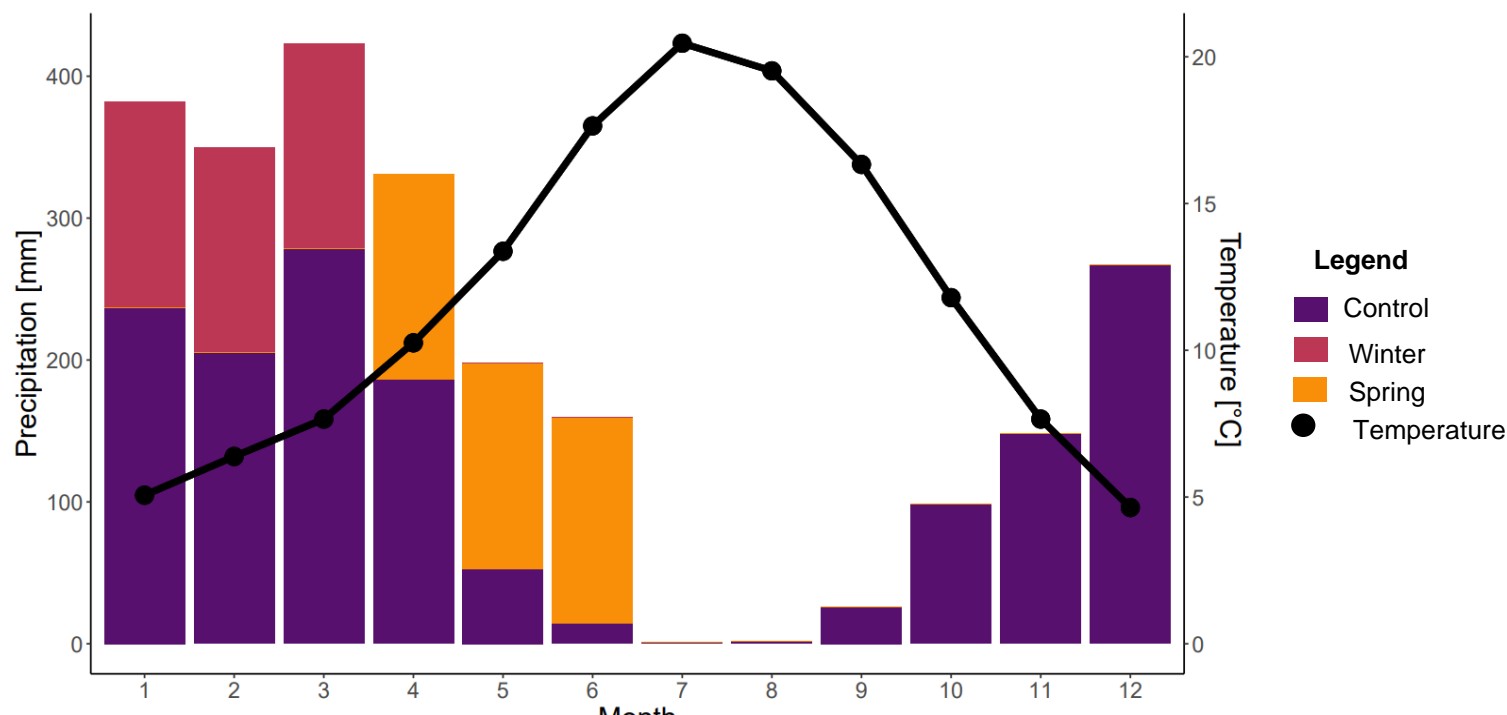

**Figure 1. Precipitation and temperature over a year at Angelo Coast Range Reserve. The control indicates ambient precipitation, whereas the added precipitation for the winter and spring treatments is shown in the months that it is added. Data was sourced from Dendra (a cyber-infrastructure project for real time data storage) for Angelo from 2012-2022. Months are numbered**




**Table 1. Functional group assignments for the bands of interest used to evaluate DRIFT spectra (based on Mainka et al. 2022)**

| Functional Group | SOM type | Wavenumber center (range) cm⁻¹ | |
| --- | --- | --- | --- |
| Aliphatic C-H stretch | Simple Plant Matter | 2925 (2976-2998) | 400 |
| | | 2850 (2870-2839) | |
| Aromatic C=C stretch | Complex Plant Matter | 1525 (1550-1500) | 405 |
| Amide, quinone, ketone C=O stretch, aromatic C=C, and/or carboxylate C-O stretch | Microbially associated OM | 1620 (1660-1580) | 410 |








**Figure 2 (a-h). Physical and chemical parameters for all treatments across the depth profile. All data are shown as means with standard error (n =3 for each treatment). Panel a shows Bulk density in g/cm³ for all treatments, panels b and c show C(%) and N(%) respectively. Panel d shows C:N ratios. Panels e and f show $\delta^{13}C$ and $\delta^{15}N$ stable isotope values. Panels g and h show pH in $H_2O$ and $CaCl_2$ respectively.**




**Table 2. Calculated cumulative carbon stocks for each treatment to examine how total carbon stocks might be changing with precipitation addition. The winter and control treatments had the greatest cumulative carbon stocks based on three cores of 0-300cm per treatment (n=9);standard error is shown in parentheses.**

| Treatment | Cumulative carbon stock (0-300cm) (g/cm$^2$) |
|---|---|
| Control | 191.2 (36.7) |
| Winter | 200.5 (34.5) |
| Spring | 171.4 (13.7) |











**Figure 3. Carbon stocks throughout 0-300 cm cores across control treatment as well as winter and spring precipitation additions. The average calculated carbon stocks in A) 50 cm depth increments with standard error and B) inset includes higher resolution for the top 50cm with 10cm increments.**





**Figure 4. Linear regressions of C:N and C stock for all depths and treatments reveal differences in nutrient dynamics between spring treatments. All linear regressions are significant but the winter treatment slope is more shallow than the control and spring treatments, which have similar slopes and higher C:N at the surface.**



**Figure 5 (a-c) . DRIFTS spectra across treatments and depths labelled with wavenumbers of interest: aliphatic compounds and simple plant matter (2976-2998 cm⁻¹ and 2870-2839 cm⁻¹); aromatic compounds and complex plant matter (1550-1500 cm⁻¹); amide, quinone, ketone stretch, aromatic and/or carboxylate stretch, and microbially associated OM (1660-1580 cm⁻¹). The Control, winter, and spring treatments are shown in panels A, B, and C, respectively, and colors represent the depth gradient.**

540





**Figure 6 (a-e). Proportions of integrated area for areas of interest in DRIFTS data, which indicate the dominance of microbially associated OM across treatments and depths. Areas of interest on DRIFTS spectra include simple plant derived functional groups (2976-2998 cm$^{-1}$ and 2870-2839 cm$^{-1}$), complex plant derived functional groups (1550-1500 cm$^{-1}$), and microbially associated OM (1660-1580 cm$^{-}$). The proportional area of interest for each 10cm depth interval for the A) control treatment, B) winter treatment, and C) spring treatments. The ratios of D) simple plant matter to microbial plant matter and E) complex plant matter to microbial plant matter by depth is also shown with averages and standard error for each 10 cm depth interval.**





575 **Table 3 Results from GAMM models for predicting δ$^{15}$N (a) and predicting SOC (b)**

| Model and terms | Estimate | z value | P value | AICc | Adj R$^2$ |
|---|---|---|---|---|---|
| **a)** | | | | | |
| **Model 1: abiotic & biotic factors** | | | | | |
| δ$^{15}$N | 1.56 | 29.35 | <2e-16 | 742.4 | 0.133 |
| Depth (Control) | | | <2e-16 *** | | |
| Depth (Winter) | | | 1.55e-06 *** | | |
| Depth (Spring) | | | 2.98e-06 *** | | |
| simple plant: microbial (Control) | | | 0.34 | | |
| simple plant: microbial (Winter) | | | 0.0076 ** | | |
| simple plant: microbial (Spring) | | | 0.31 | | |
| complex plant: microbial (Control) | | | 0.48 | | |
| complex plant: microbial (Winter) | | | 0.0023 ** | | |
| complex plant: microbial (Spring) | | | 0.019 ** | | |
| **Model 2: abiotic factors** | | | | | |
| δ$^{15}$N | 1.56 | 30.38 | <2e-16 | 747.9 | 0.101 |
| Depth (Control) | | | <2e-16 *** | | |
| Depth (Winter) | | | 4.04e-05 *** | | |
| Depth (Spring) | | | 6.4e-04 *** | | |
| **b)** | | | | | |
| **Model 1: abiotic & biotic factors** | | | | | |
| log (SOC) | -1.44 | -26.3 | <2e-16 | 151.9 | 0.722 |
| Depth (Control) | | | <2e-16*** | | |
| Depth (Winter) | | | <2e-16*** | | |
| Depth (Spring) | | | <2e-16*** | | |
| simple plant: microbial (Control) | | | 0.25 | | |
| simple plant: microbial (Winter) | | | 0.088 | | |
| simple plant: microbial (Spring) | | | 0.096 | | |
| complex plant: microbial (Control) | | | 0.078 | | |
| complex plant: microbial (Winter) | | | 0.13 | | |
| complex plant: microbial (Spring) | | | 0.69 | | |
| **Model 2: abiotic factors** | | | | | |
| log (SOC) | -1.46 | -25.6 | <2e-16*** | 153.9 | 0.725 |
| Depth (Control) | | | <2e-16*** | | |
| Depth (Winter) | | | <2e-16*** | | |
| Depth (Spring) | | | <2e-16*** | | |



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
