# Peer review of "Carbon and Nitrogen Dynamics in Subsoils After 20 years of Added Precipitation in a Mediterranean Grassland"

_EGUsphere, 2024_

## Author Response (AR1)

**Responses to reviewer 1:**

**General Comments:**

This study evaluated soil carbon (C) and nitrogen (N) concentrations, stocks, and isotopes, as well as soil functional groups in the full soil profile (3m depth) under 20 years of altered precipitation regimes (increased spring or winter precipitation). The research finds relatively minimal responses of these response variables to their treatments but find slightly higher subsoil C stocks in the treatment with higher winter precipitation and suggest this might be related to greater plant inputs. I think this paper uses an impressive dataset to get at an important and understudied question. However, I find some of the logic in this paper, particularly in the discussion, a bit difficult to follow, and suggest some areas where it could be clarified below. Overall, I think this paper is a useful addition to our understanding of subsoil biogeochemical responses to climate change but would benefit from some clarification.

Response: We would like to thank this reviewer for their thoughtful comments. We have done our best to apply them, and they have already greatly improved the manuscript and its clarity.

**Specific Comments:**

Lines 8-11: I find having these sentences about previous studies in the abstract a bit confusing. I think introducing the previous studies in the introduction is sufficient and these lines could be removed from the abstract.

Response: Agreed, they have been removed!

Lines 45-47: Please add citation(s) for this sentence.

Response: We have added two citations in line 43 to address this:

Min, K., Slessarev, E., Kan, M., McFarlane, K., Oerter, E., Pett-Ridge, J., Nuccio, E., and Berhe, A. A.: Active microbial biomass decreases, but microbial growth potential remains similar across soil depth profiles under deeply-vs. shallow-rooted plants, Soil Biol. Biochem., 162, 108401, https://doi.org/10.1016/j.soilbio.2021.108401, 2021

Jilling, A., Keiluweit, M., Contosta, A. R., Frey, S., Schimel, J., Schnecker, J., Smith, R. G., Tiemann, L., and Grandy, A. S.: Minerals in the rhizosphere: overlooked mediators of soil nitrogen availability to plants and microbes, Biogeochemistry, 139, 103–122, https://doi.org/10.1007/s10533-018-0459-5, 2018

Lines 47-50: I am a bit confused by this sentence – it seems to suggest that Fe/Al oxyhydroxides are broadly concentrated in deep soils, but I don't believe the cited work provides evidence for that. I agree that Fe/Al oxyhydroxides are important for soil C stabilization, but I believe the point the authors are trying to make are just that there are generally more minerals at depth and thus greater surface area for sorption. Could the authors please clarify here?

Response: The citations in this sentence are largely addressing the importance of mineral associations in stabilization carbon compounds, but yes, we wanted to make the point of greater surface area of these oxides at depth and how consequential that is for carbon. We clarified the sentence structure to better communicate these ideas, and added specific citations for the evidence of greater surface area of mineral phases in subsoils horizons in lines 44-46:

"Deep soils often have a greater surface area of mineral surfaces, such as Fe/Al oxyhydroxides (Mikutta et al., 2006; Rumpel and Kögel-Knabner, 2011). These mineral surfaces are important for carbon stabilization, and allow for protective and stabilizing associations for carbon compounds that would otherwise be quickly decomposed (Kleber et al., 2005; Porras et al., 2017; Schmidt et al., 2011). "

Lines 78-81: Could the authors briefly discuss the findings of the previous studies at this site and how those findings might inform this study?

Response: I added a paragraph to address this comment starting at line (76). I began by adding this information to the previous paragraph, regarding other precipitation experiments, but felt that the paragraph became overloaded and too long. This way, the introduction becomes increasingly specific (other long term experiments -> the angelo experiment and previous work done there -> our hypotheses for this study) and addresses your comment. A lot of the studies that I mention in this paragraph are also brought back up in the discussion as well. This paragraph now reads (lines 76-93):

"Previous work at the Angelo experiment has suggested important biotic feedbacks, both plant and microbial, with changing precipitation amount and seasonality. The Angelo experiment has been ongoing for 20 years, and is testing the impacts of increased precipitation combined with changing seasonality. More specifically, it is testing the impact of shifting seasonality of precipitation to the spring months (Mar-June) in a Mediterranean climate where most of the precipitation for the water year typically takes place in the winter months (Nov-Feb). This site had multiple studies occur at the 6 and 10 year mark of the experiment (Berhe et al., 2012; Cruz-Martínez et al., 2012; Hawkes et al., 2011; Suttle et al., 2007), but this study represents one of the first long term follow ups on that experiment to great depth. Previous work at the Angelo experiment has suggested that plant and fungal communities are especially sensitive to changing amount and seasonality of precipitation. More specifically, it was found that added spring precipitation caused reductions in plant diversity (Suttle et al., 2007), while fungal communities were less diverse under both winter and spring additions (Hawkes et al., 2011). Microbial communities, on the other hand, were relatively more robust and resilient to changing water conditions (Cruz-Martínez et al., 2009), however, a later study suggested that longer term studies might be missing important short term variation driven in part by rainfall fluctuation (Cruz-Martínez et al., 2012). More recent work on microbial communities at the Angelo experiment suggested that extended rainfall decreased depth based differentiation in microbial community composition (Diamond et al., 2018). Seven years of changing seasonality and amount of precipitation also has been found to affect soil biogeochemical processes, with increased winter precipitation diminishing the role of Fe/Al oxyhydroxides in C stabilization (Berhe et al., 2012). While significant work has been done at Angelo on biotic responses to changing precipitation regimes, less work has been done on soil biogeochemistry, especially at depth (>50cm). "

Paragraph at line 81: I am not sure this information should be in the introduction – it feels more appropriate for the methods section in my opinion.

Response: This information has been moved to section 2.4 in the methods (paragraph starting at line 148)

Line 96-97: I wonder if it would be good to bring up this seasonality change earlier in the introduction? I think it is a unique aspect of this study that could be used to distinguish it from other studies and so it might make sense to bring it up at the end of the paragraph starting at line 65.

Response: Text regarding the specifics of the experiment have been moved up in the introduction (lines 77-80).

Line 102: Would the authors be able to provide their hypothesized responses here? The introduction is informative and it seems like it could inform some hypotheses.

Response: We added hypotheses from lines 99-103 in the last paragraph of the introduction, and informed them based on previous work done at the Angelo experiment. These added lines read as follows:

"Our hypotheses were largely based on a previous study at the precipitation experiment which focused on belowground processes (Berhe et al., 2012). This study found lower C concentration with added winter precipitation, and higher C concentration with added spring precipitation. Authors also found an accumulation of easily assimilated and less decomposed SOM and reduced rates of decomposition with spring addition. Based on this, we hypothesized that (a) there would be a reduction in C stocks with winter addition. We also hypothesized that there would be an (b) accumulation of aliphatic functional groups with spring addition and (c) a stronger association between inputs and carbon stocks with spring addition."

Section 2.3: At some point in this section, it would be helpful to enumerate the total amount of samples – this seems like it was a lot of work and that should be emphasized! This would also provide clarity for the reader.

Response: This has been added in line 133. It was indeed a lot of collected soil samples!

Line 179: Please add an R citation.

Response: This has been added in line 193

Line 180: Please clarify which depth distributions were used for which analyses.

Response: This sentence is referring to the statistics for the carbon stock calculations, and the clarification has been added to the sentence (line 193-194) to further help the reader and reads:

"Differences between treatments for C stocks were evaluated through Kruskal-Wallis test within each 10cm (Fig. 3b) or 50cm (Fig. 3a) depth interval depending on the analysis."

Line 193: If the full model with all predictors had equivalent or smaller AIC than the depth model, why include the depth model as a separate analysis?

Response: We decided to keep in the depth model due to its high explanatory power in its own right. Depth is an important factor when considering any aspect of soil C processes, so we wanted to acknowledge that with the model structure, especially given that this study goes to 3m.

Section 3.1: Could the authors add some quantitative data here? These could be ranges of the physical and elemental properties or by what percent treatments increased or decreased these properties.

Response: We added more quantitative data and specific references to data ranges for bulk density, C%, and C:N values throughout section 3.1..

Paragraph at line 221: Could the authors clarify what they mean by "gross changes (cumulative C stocks)"? Are these values different than measured C stocks? If not, I would suggest using simpler

phrasing. Gross changes sound like there is a comparison over time or across treatments. Additionally, when data are presented for differences at 150cm (or at 300cm), is that for the slice of soil from 140-150 or 150-160cm or something different? Please clarify in the text.

Response: Agreed, the language here was unclear. The intent of this was to reinforce the first sentence of section 3.2, in which we state we found no statistical differences between carbon stock estimates. We have eliminated the language regarding gross changes in this sentence, and have rather emphasized that we are discussing trends of carbon stocks due to the lack of statistical difference. This now reads (lines 235-236):

"All discussion of these results in this paragraph refer to trends between calculated cumulative stocks and across depth profiles due to the lack of statistical significance determined by the Kruskal-Wallis tests."

Paragraph at line 231: I am not sure how soil C:N is meant to be indicative of plant inputs – could the authors cite something to support this? Soil C:N is influenced by other processes beyond plant inputs and so I don't understand the presumed connection here. I think this analysis and paragraph could be removed given the DRIFTS analysis more directly addresses potential inputs in my opinion.

Response: I do agree that C:N integrates many processes, but I also think that the C:N values at this site show interesting patterns throughout the profile. The surface C:N values in the control are >20, while subsoil C:N was incredibly low (<5). There are citations to support that these high C:N values at the surface might be reflective of grassland plant tissue (Bell et al., 2014; Nierop et al., 2001). It has also been shown that microbial material has a low C:N (Dijkstra et al., 2006) and that microbial byproducts, like amino sugars and amino acids, can decrease C:N ratios (Knicker, 2011). I do want to also acknowledge that C:N is related to decomposition processes (Conen et al., 2007, 2013). I have added these citations and justifications to the text, but am open to also removing this analysis if this still doesn't seem like good enough justification. We have added text to lines 247-251 which read:

"These is evidence to support that high C:N values are indicative of plant tissues (Bell et al., 2014; Nierop et al., 2001), while low C:N values are indicative of microbial byproducts such as amino acids and amino sugars (Knicker, 2011), Overall, C:N values integrate multiple processes that can be difficult to disentangle, such the balance between inputs and decomposition (Conen et al., 2007)."

Paragraph at line 248: I suggest the authors remove the linear regression results from this paragraph. Only the GAMMS were introduced in the methods, so I think having linear regression results here are confusing for the reader. Additionally, I think it would be useful to comment on the directionality of the relationships with the GAMMS. Given this data is a notable discussion point, the authors could also consider moving these plots to the main text.

Response: We have removed the linear regression text from the results paragraph, and added note of the directionality of relationships where relevant in lines 273-274. We have gone back and forth about the inclusion of the GAMM plots between authors and others who have reviewed the text, and we landed on the table better communicating the relevant statistics and important explanatory variables.

Lines 280-281: I see the authors cite von Haden here – did they consider analyzing the soil C stocks using the equivalent soil mass approach to see if that influenced their results given the slightly variable bulk density between treatments?

Response: Yes, the equivalent soil mass approach was considered by authors for this dataset. However, ESM requires continuous cores, and there were a couple of damaged samples here and there that created discontinuities that would have greatly reduced statistical power in this case (which is already pretty limited with the small n we are working with). For this reason, we decided against applying this method to our dataset.

Lines 281-284: I am confused by this sentence – I thought this work found microbial inputs as dominant throughout the soil profile?

Response: I removed this sentence as I think it was unclear and the patterns of C and functional groups with depth are explained with more detail later in the paragraph. The DRIFTS data did show a significant proportion of microbial functional groups (~50%), but was roughly equivalent if simple and complex plant matter are taken together.

Lines 292-305: I am not quite following the reasoning in this paragraph. Is there work demonstrating that increased precipitation is likely to lead to increased root exudation? If so, could that be referenced here? Additionally, it could be helpful to start from the findings and use those to build a rationale. Specifically, more clearly explaining how relationships between %C or $\delta^{15}N$ and the DRIFTS ratios support the reasoning in this paragraph would be helpful.

Response: We have significantly altered the structure of section 4.1 to try to improve the reasoning and explanation of rationale. We moved explanation of results up (to better start from findings, lines 294-301), and added more citations regarding the relationship between root exudation rates and precipitation.

Line 314: I am missing the connection here – were there more invasive plants in the spring treatment? Pleas clarify in the text.

Response: Yes, by the measurements of Suttle et al. (2007), they found greater annual grass biomass in the spring and winter treatments and overall less plant richness. They interpreted this, along with decreased invertebrate richness in the spring plots, as an overall simplification in the biodiversity and food web with spring addition of rainfall. We tried to more explicitly make the connection between reduced diversity and annual grasses with the addition of the following sentence (line 305-306):

"More specifically, Suttle et al. (2007) found that annual grass biomass was greater and plant richness lower with spring addition."

Lines 334-335: Is this sentence meaning that the authors lack plant $^{13}C$ data? Could they explain whether the soil $^{13}C$ data fit the proposed theory of more plant inputs in the spring and winter treatments (e.g., whether $^{13}C$ is lower in soil profiles in those treatments)?

Response: We clarified the language in this sentence to address the lack of plant tissue d13C data as well as added a couple of sentences regarding soil isotopic values. I do think we are seeing this pattern in the soil data, but it is not statistically significant and very slight. The revised and added sentences read as follows (line 351-356):

"Specifically, we think that formed carbon in plant inputs is lowered in $\delta^{13}C$ value, and is could be driving down $\delta^{13}C$ values in the soil profile. However, we lack isotopic measurements of plant tissue and think it would be a valuable avenue for future work. Soil $\delta^{13}C$ values in the Spring and Winter treatments were

not significantly different from the Control, and the high variability in this measurement makes it hard to detect if we are seeing this pattern in our data, however there is some evidence of slightly lower $\delta^{13}C$ values in the spring and winter treatments from 0-100cm and again from 120-200cm (Fig. 2e)."

Paragraph at line 336: Could the authors connect the information in this paragraph (which I think is useful!) to the findings of this study?

Response: We addressed this comment by adding sentenced to the end of this paragraph about the impacts this could have on potential subsoil carbon sequestration (lines 367-370):

"While this study lacked explicit measurement of microbial communities with precipitation addition, we did observe increased translocation of C throughout the depth profile with precipitation addition. This ability of subsoil microbial communities to quickly take advantage of fresh C inputs could affect the long-term sequestration potential of subsoils affected by increased precipitation."

Line 366-367: Again, I am a little confused by this focus on altered plant inputs – perhaps clarifying this in the discussion would help clarify why this finding should be included in the conclusion.

Response: We also altered this sentence to better acknowledge that inputs to subsoils could be microbial and plant derived. It now reads (lines 374-375):

"The addition of carbon (both microbial and plant derived) in subsoils in the winter treatment could fundamentally alter C cycling in subsoils"

**Technical Corrections:**

Line 31: I believe the authors might mean "not fully accounted".

Response: Edited

Throughout: Please use C instead of carbon once defined.

Response: Addressed

Line 239: There seems to be some extra text here.

Response: This extra text has been deleted

Line 272: Do the authors mean precipitation addition and not carbon addition here?

Response: Yes, we meant precipitation addition!

**Responses to reviewer 2:**

General comment:

The study provides novel insights on carbon dynamics in subsoils as it analyzes 3m depth soil profiles that underwent a 20-yr rainfall manipulation experiment that considers the effect of seasonality (more spring or winter precipitation) on (subsoil) C dynamics. The manuscript is well written overall and answers the research questions using valid methods. However, the presentation and discussion of the results could in parts be clearer and more specific regarding the mechanisms underlying their data. I would recommend this paper for publication in Biogeosciences given that the authors clarify their presentation of results and discussion.

Response: We would like to thank this reviewer for their comments! We have applied them and they have especially clarified the methods.

Specific comments:

Section 2.1: In my opinion, it would be valuable to add information on the landscape position of the experimental site. Is there a potential influence of alluvial sediments along the soil profiles?

Response: This site overlays a fork of the Eel river, and it's unlikely that alluvial sediments are being deposited onto the soil profile. We have added more on this in lines 113-114 which reads:

"The parent material is largely graywacke and mudstone, and derived from Cretaceous marine grey-wacke sandstones and mudstones of the Franciscan complex, and the site overlays a bedrock terrace of the South Fork Eel River (Berhe et al., 2012)."

l. 142-43: The sentence on bulk density calculation could be reformulated as you probably used the dry mass of the < 2 mm fraction (corrected for the > 2 mm fraction) to calculate BD.

Response: Corrected to <2mm fraction!

l.173: Could you maybe add a sentence on how areas under the curve were calculated and which software you used? Please add also a sentence on how the total area under the curve is defined.

Response: We added more information regarding how the area under the curve was calculated and defined in lines 190-193 that read:

"We integrated the area under the curve in R (v4.2.1, R Core Team, 2022) for our functional groups of interest (aliphatic, aromatic, and amide). Bounds for integration are reported in Table 1 as "range." We then normalized the area under the curve for our functional groups of interest to 100%."

l. 183: It would be good to add a description of the linear regressions that were used e.g. in Figure 4.

Response: We added sentences in lines 206-210 to address this, which read:

"To better understand relationships between C stock and inputs, we performed a linear regression on C stock and C:N values. While C:N values integrate many processes, there is evidence to suggest that higher C:N (>20) indicates plant inputs, while low C:N values (<8) indicate microbial inputs (Bell et al., 2014; Knicker, 2011; Nierop et al., 2001). We interpreted a high C:N as indicative of more plant inputs, while a low C:N would be indicative of more microbial inputs and decomposition."

l. 192-94: Did you check for autocorrelation among predictor variables, e.g. using variance inflation factors?

Response: Variance inflation factors (VIFs) measure collinearity, which is a valid concern, but measuring VIF of a GAM is not an appropriate test due to the smoothing functions present in a GAM. The extension of collinearity to GAMs is concurvity, which we did check. There was minimal concurvity in the carbon models, and some concurvity in the $\delta^{15}N$ models. However, GAMs using the mgcv package are still robust even with concurvity (Wood, 2008). We added a line in the methods to address this, which reads:

"We also checked concurvity (the exnteion of collinearity to a GAMM) through the mgcv package."

Citation: Wood, Simon N. 2008. "Fast Stable Direct Fitting and Smoothness Selection for Generalized Additive Models." Journal of the Royal Statistical Society: Series B (Statistical Methodology) 70 (3): 495–518

l. 241-42: Is it the proportional contribution relative to the total area under the curve? This could be clarified.

Response:

We have added text to the methods to clarify this (see comment regarding lines 190-193) but have also added additional text here in lines 277-278 to reiterate:

"We calculated proportional relationships by normalizing the area under the curve for our functional groups of interest to 100%."

l. 259-60: Maybe you could be clearer here and specify the unique relationship to the reader.

Response: We agree that the language could be clearer here. This sentence has been edited to read (line 194):

"Through visualizing our GAMM, we saw significant negative relationships in the winter and spring treatment for the relationship between the simple plant: microbial and $\delta^{15}N$ values (Fig. S2a)."

l. 280-81: Could you please also provide the reason(s) why significant variability is introduced from fixed depth bulk density measurements?

Response:

We added more detail to this sentence, but the main reason would be if bulk density (or specifically soil volume) were changing over time. This sentence now reads (line 317):

"This could be due to significant variability introduced from fixed depth bulk density measurements due to changing soil volume over time (von Haden et al., 2020). "

l. 295-305: I think the whole paragraph might benefit from more specific wording. "Unique relationships" are very unspecific and make it harder to follow the authors' explanations in this section but also the results section of the paper (see for instance l. 259). It might be recommendable to more explicitly mention the actual trends, e.g. that a wider C:N ratio indicates more plant-derived C etc.

Response: Based on comments also from reviewer 1, we have edited this paragraph for greater clarity and have significantly altered the structure of section 4.1 to try to improve the reasoning and explanation of rationale. We moved explanation of results up (to better start from findings, lines 294-301), and added more citations regarding the relationship between root exudation rates and precipitation. This section now reads:

"We observed clear signals of greater C translocation throughout the soil profile as a result of added precipitation, but where this C accumulated was based on seasonality. Added precipitation seemed to lead to greater transport of C throughout the profile in the winter and spring treatment especially. However, we did not see any statistically significant differences in C stocks across both the subsoil and surface C stock measurements (Figure 3a-3b). This could be due to significant variability introduced from fixed depth bulk density measurements due to changing soil volume over time (von Haden et al., 2020).  We observed significant evidence for slightly higher C in the winter treatment being plant derived based on both C:N and DRIFTS data. We also saw evidence of greater biotic processing in the Spring treatment when comparing $d^{15}N$ to the simple plant matter: microbially associated DRIFTS ratio (Table 3) and the overall lower and more constrained C:N values (Fig. 4).  We also saw positive and significant trends when we related C concentrations to the ratio of complex plant matter to microbial associated OM (Fig. S2b), highlighting a unique relationship between complex plant inputs and C especially in the winter and spring that may be related to biotic  inputs. It is important to note that evidence of subsoil C accumulation in the winter treatment was only made possible by incorporating measurements from deeper than 1m, and that depth was a significant predictor in our GAMM models across all treatments (Table 3). Previous work suggests that the main sources of organic matter into subsoils are plant derived compounds (roots and root exudates), dissolved organic matter (DOM) and bioturbation (Rumpel and Kögel-Knabner, 2011). Work at the Angelo experiment has shown evidence of changing rooting patterns and greater overall biomass with increased precipitation at this site (Suttle et al., 2007). Rooting depth and root exudation rates are affected by availability of soil water, and direction of response is strongly species dependent (Li et al., 2021; Souza et al., 2023; Staszel et al., 2022). However, it has been found through $^{13}CO_2$ pulse labelling studies that drought reduces transfer of newly fixed C to microbes (Fuchslueger et al., 2014; Ruehr et al., 2009), while manipulative experiments with rainfall addition increase root C exudation rates (Li et al., 2021). A recent meta-analysis noted that decreased precipitation also slows the belowground C cycle, while precipitation increases promote nearly every aspect, such as C stock, substrate supply, microbial activity, and respiration (Abbasi et al., 2020). This is due to interactions between precipitation and biological entities, namely plants and microbes. Increased precipitation root respiration and belowground NPP are positively correlated with soil water availability, and enhance plant growth and photosynthetic rates (Heisler-White et al., 2008; Maire et al., 2015). Wetting of dry soil also has a dramatic impact on soil

microbes due to increased substrate availability and reactivation of dormant microbes, yielding respiration pulses known as the Birch effect (Salazar et al., 2018; Schimel et al., 2007; Skopp et al., 1990). Overall, greater precipitation in the winter could contribute to greater root exudation in surface soils that then gets quickly fixed by soil microbes. Our results show that in the winter treatment plots, the additional precipitation in the already wet winter season likely increases root exudation, where this increased C input coincides with lower temperatures and lesser biological activity in soil. Whereas in the spring treatment, soil C is exposed to greater microbial processing. Thus, there is greater movement of this plant derived inputs moving down the profile in both the spring and winter treatments, but to greater depths in the spring treatment, likely as DOM. Recent plot-scale studies have proposed that OM formation in subsoils is linked to a complex cascade model, in which OM is sorbed, microbially processed, and remobilized in cycles as it migrates down the profile (Liebmann et al., 2020). "

l. 365-369: It is just a suggestion but, in my opinion, the last sentences could benefit from highlighting the importance of timing of precipitation and the evidence of translocation of C.

Response: We agree that the final sentence of this study could pack more of a punch and better highlight the results from the study. To address this, we added two sentences to the end of the manuscript:

"This study found that increased precipitation has the potential to increase C translocation to deeper layers, and that added winter precipitation causes a shift towards more abiotic control on soil C cycling. Changing precipitation amount and seasonality need to be taken into account when considering the C sequestration potential of Mediterranean grasslands."

Technical comments:

l. 37: It should be mm instead of cm (2000 – 3000 mm).

Response: This has been corrected to mm

l. 78: delete "was" in "This site was had […]"

Response: Deleted!

l. 84-85: correct "have to remain close more" to "[…] more closed"

Response: Corrected

l. 146: typo in "sieve[d] to 2mm]"

Response: Corrected

l. 182-83: There is a repetition, and it would be good to add a citation for R.

Response: A citation for R has been added, and redundant language replaced

l. 270: delete "is" before "varies"

Response: Deleted!

---

## Author Response (AR2)

Author response to Reviewer 1:

**Line 347: Should the parentheses by +/- rather than just - ?**

Author response:

While there was not a $\pm$ in that line, I believe that reviewer 1's comment had to do with line 26, which initially read:

"Current estimates of global SOC stocks to 1m have converged to 1100-1500 Pg, but estimates to 3m depths are 2800 Pg ($\pm$ 700 Pg) (Jackson et al., 2017)."

Now it reads:

"Current estimates of global SOC stocks to 1m have converged to 1100-1500 Pg, but estimates to 3m depths are 2800 Pg$\pm$700 Pg (Jackson et al., 2017)."

I also double checked throughout the ms for other uses of parentheses around $\pm$ to make sure notation was correct.